# Intermittent low-dose far-UVC irradiation inhibits growth of common mold below threshold limit value

Emilie Hage Mogensen[1]*, Christian Kanstrup Holm[1,2]

1 UV Medico A/S, Aarhus, Denmark, 2 Department of Biomedicine, Aarhus University, Aarhus, Denmark

* emo@uvmedico.com

## Abstract

Mold infestations in buildings pose significant challenges to human health, affecting both private residences and hospitals. While molds commonly trigger asthma and allergies in the immunocompetent, they can cause life-threatening diseases in the immunocompromised. Currently, there is an unmet need for new strategies to reduce or prevent mold infestations. Far-UVC technology can inactivate microorganisms while remaining safe for humans. This study investigates the inhibitory efficacy of far-UVC light at 222 nm on the growth of common mold-producing fungi, specifically *Penicillium candidum*, when delivered in low-dose on-off duty cycles, a configuration consistent with its use in real-world settings. The inhibitory effect of the low-dose duty cycles was assessed on growth induced by i) an adjacent spore-producing *P. candidum* donor and ii) *P. candidum* spores seeded directly onto agar plates. In both setups, the far-UVC light significantly inhibited both vertical and horizontal growth of *P. candidum*, even when the UV doses were below the Threshold Value Limit of 23 mJ/cm$^2$. These results suggest that far-UVC light holds the potential to improve indoor air quality by reducing or preventing mold growth, also when people are present.

## Introduction

Fungi can be found both indoors and outdoors, and they thrive in a broad range of conditions, spanning diverse temperatures. High humidity and water activity are prerequisites for their optimal growth [1]. Filamentous fungi can form spore-producing molds, and typically, outdoor levels of fungal spores exceed indoor levels [2], unless there is a source such as wall or floor mold patches [3]. A study conducted on buildings exposed to prolonged water leaks and increased humidity showed that *Penicillium* species were the most common species isolated from water-damaged materials, followed by *Aspergillus* and *Chaetomium* as the subsequent most commonly identified species [4]. Mold growth can lead to significant structural damage to buildings when it decomposes organic matter such as wood, wallpaper, cardboard, and linoleum [4].

When considering air quality, indoor mold or fungal growth can pose a significant challenge. The sick building syndrome encompasses various symptoms such as headaches, a blocked or runny nose, dry and itchy skin, as well as coughing. These symptoms manifest

**Data Availability Statement:** All relevant data are within the manuscript.

**Funding:** EHM and CKH received salary from UV Medico A/S, the funder of the study. The specific roles of these authors are articulated in the 'authors

contributions' section. The funder provided support in the form of salaries for authors, but did not play any role in study design, data collection and analysis, decision to publish, or preparation of the manuscript.

**Competing interests:** EHM is a full-time employee and CKH is a co-founder and an employee at UV Medico A/S. This does not alter our adherence to PLOS ONE policies on sharing data and materials. There are no patents, products in development, or marketed products associated with this research to declare.

when susceptible individuals are inside specific buildings but tend to decrease after spending time outside of them. Different fungal species, including *Penicillium* [5, 6], have been associated with sick building syndrome symptoms. *Penicillium* is classified as Deuteromycetes, the same class as most other molds found in sick buildings [7], and exposure to *Penicillium* spp. in indoor environments can increase the susceptibility to asthma [8] and allergies [6, 9] in certain populations. In hospitals, mold outbreaks can cause increased morbidity and mortality, particularly for immunocompromised patients [10]. To protect public health and avoid these adverse health effects, the World Health Organization recommends preventing or limiting microbial growth on indoor surfaces [11].

UVC light has been recognized for its germicidal efficacy for a century. Typically, mercury vapor light sources emitting light at 254 nm are used, but several precautions must be taken, as this light can damage skin and eyes. Within the UVC spectrum, far-UVC is defined to be within the 200–230 nm wavelength range, which includes krypton chloride excimer lamps that emit light at 222 nm. The KrCl lamps often contain filters to ensure that harmful wavelengths outside 222 nm are inhibited. Despite having a wavelength only 32 nm from conventional UV light at 254 nm, the properties of 222 nm are fundamentally different because of its high absorption in proteins [12]. Studies have demonstrated that far-UVC light can effectively inactivate microorganisms, including bacteria and viruses [12–15], while remaining safe for human and rodent skin [16–23] and eyes [24, 25]. The Threshold Limit Value (TLV) set by the International Commission on Non-Ionising Radiation Protection is 23 mJ/cm$^2$ [26], which represents the maximum allowable irradiation to which a worker can be exposed during an 8-hour working day. However, safety studies have led the American Conference of Governmental Industrial Hygienists to recommend a new TLV of 161 mJ/cm$^2$ for the eyes and of 479 mJ/cm$^2$ for the skin [27].

These TLVs apply to healthy adults as the effects on children and skin patients is yet to be studied.

Studies have also demonstrated the ability of far-UVC light at 222 nm to inactivate various fungal species [12, 13, 28, 29], typically by using a single high dose for inactivation. However, in real-world settings, far-UVC lamps are often configured to cycle on and off, delivering intermittent low-intensity doses throughout the day. This approach keeps the UV dose, to which individuals are exposed, below the TLV. As a result, microorganisms experience intermittent low doses of far-UVC light instead of one large dose. Here we report how cycling low doses of far-UVC light inhibits the growth of *Penicillium candidum* on agar plates, both when it stems from airborne spores and when seeded directly onto agar plates.

## Material and methods

### Far-UVC light source

The far-UVC source used in this study was a krypton chloride excimer lamp (UV222$^{TM}$, UV Medico, Denmark), equipped with a filtered light source emitting light at 222 nm (Care222, Ushio Inc., Japan). The filter blocked emissions outside the 222 nm peak. The lamp had an output of 120 mW, an emission angle of 60 degrees, and an intensity of 13.7 μW/cm$^2$ at a 1-meter distance from the lamp, measured using a UIT2400 handheld light meter for 222 nm (Ushio Inc., Japan). The lamp was equipped with software (UV Medico, Denmark) that allowed for duty cycling configuration.

### Inhibition of *P. candidum* growth from airborne spores by far-UVC light

Two identical boxes (80 x 120 x 185 cm) were constructed, each equipped with a far-UVC lamp affixed to the ceiling. Air movement was ensured by the installation of a ventilation fan

in one corner of each box. To maintain a constant humidity level of 80%, humidifiers (CR 7952 Camry, Adler, Poland) were employed. Preparation of donor plates was done by inoculating Dichloran-Glycerol 18% agar plates (DG18 LAB-AGAR™, Biomaxima, Poland) with *P. candidum* (SWING FD PCA-1 culture, Chr. Hansen, Denmark) and letting them incubate at room temperature until the mycelium completely covered the agar surface. Subsequently, the donor plates were transferred to the boxes, one donor plate in each box, accompanied by three to four sterile DG18 plates placed adjacent to each donor plate. The plates were left in the box for one to two weeks to allow the spores to induce growth on the adjacent agar plates.

The far-UVC lamp was positioned 178 cm away from the agar plates. At this distance, it delivered an intensity of 4.4 $\mu W/cm^2$ to the agar plates and was programmed to cycle on and off. The total accumulated UV doses and corresponding on/off cycles per 24 hours were as follows: 25 $mJ/cm^2$ (1 minute on and 14 minutes off), 50 $mJ/cm^2$ (2 minutes on and 12 minutes off), and 100 $mJ/cm^2$ (4 minutes on and 11 minutes off). In one box, the lamp remained turned off all the time to serve as a control condition, and the control and far-UVC boxes were swapped each time the experiment was repeated. Multiple UV doses were screened once, and the minimum dose (25 $mJ/cm^2$) was selected for additional repetitions, totaling four technical replicates at this low dose. The growth was monitored by measuring the covered area on the agar plates on the fourth day after the initial colony appeared.

### Inhibition of *P. candidum* growth from seeded spores by far-UVC light

To further investigate the impact of cycling low doses of far-UVC light on mold growth, four identical boxes (63 x 125 x 140 cm) were built with a far-UVC lamp installed in the ceiling of each box. Humidifiers (CR 7964 Camry, Adler, Poland) maintained the humidity at 80–90% within the boxes. *P. candidum* spores were dissolved in phosphate-buffered saline with a concentration of $3.4 \times 10^6$ spores/mL and transferred to DG18 agar plates in 8 $\mu L$ droplets, with six droplets on each plate. A seeded agar plate was positioned at the bottom of each box, directly under the far-UVC lamp. The distance between the lamp and the agar plate was 130 cm, resulting in an intensity of 8.16 $\mu W/cm^2$ being delivered to the agar plate. The accumulated UV doses and corresponding cycles per 24 hours were as follows: 10 $mJ/cm^2$ (13 seconds on and 14 minutes and 47 seconds off), 15 $mJ/cm^2$ (19 seconds on and 14 minutes and 41 seconds off), and 20 $mJ/cm^2$ (26 seconds on and 14 minutes and 34 seconds off). The agar plates were incubated in the boxes for seven days, and four technical replicates were conducted. The growth was monitored daily by capturing pictures for visual assessment and by measuring the increase in diameter of all seeded droplets.

### Statistical analysis

Unpaired Welch's t-tests were performed, assuming parametric data distribution. The tests were conducted to evaluate the difference in growth of *P. candidum* under control conditions versus far-UVC irradiated conditions, with the growth stemming both from airborne spores and spores seeded directly onto an agar plate. The statistical level of significance was set to 5%, and data was analyzed in GraphPad Prism version 10.0.3.

## Results

### Far-UVC light at 222 nm inhibits growth of *P. candidum*

In the initial experiment, donor agar plates covered with *P. candidum* mycelium were placed in boxes with 80% humidity, and air movement was ensured by a fan to spread airborne spores. Sterile receiver agar plates were positioned adjacently to the donor plates, a far-UVC

lamp was installed in the ceiling of each box, as depicted in Fig 1A. The total UV doses accumulated over 24-hour periods during each of three duty cycles were 100, 50, and 25 mJ/cm$^2$, respectively. As a control, the far-UVC lamp was turned off in one of the boxes. Fig 1B displays representative images of the growth on the plates, both under far-UVC irradiation (lower plates) and non-irradiated control plates (upper plates). In total, 74% of the control plates showed visible growth, with an average of 21.7% of the agar surface being covered by mycelium (Fig 1C) four days after the initial colony appeared. In contrast, none of the plates in the 100 or 50 mJ/cm$^2$ conditions showed any growth. As the lowest dose of 25 mJ/cm$^2$ exhibited a comparable inhibitory effect to the higher doses, we conducted this condition four times for verification. A third of the far-UVC-irradiated plates showed growth, but only as minor colonies, and the total surface area covered on any of the plates never exceeded 1%.

To determine if the DG18 agar plates were affected by the far-UVC light in a way that could influence *P. candidum* growth, such as light-dependent generation of reactive oxygen species [30], we pre-irradiated DG18 agar plates with 0 (control), 100, or 400 mJ/cm$^2$ far-UVC light and monitored the growth. We found no significant difference between the control and far-UVC irradiated plates (S1 Fig).

Additionally, to assess the levels of ozone generated by the far-UVC lamp, ozone measurements were performed in the box using the highest far-UVC dose of 100 mJ/cm$^2$ accumulated over 24 hours. However, the ozone level failed to reach the detection limit of 20 ppb (S2 Fig). Thus, the level did not exceed WHO's estimated background ozone level of 35.7 ppb [31].

These results demonstrate that applying far-UVC light in duty cycles can effectively inhibit the growth of *P. candidum*.

## Growth inhibition can be achieved by using doses below the current TLV

In the second experiment, we proceeded to assess the effect of duty cycles with low doses of far-UVC light below the TLV. Agar plates seeded with droplets of *P. candidum* spores in solution were placed in boxes with 80–90% humidity and with far-UVC lamps in the ceiling as depicted in Fig 2A. The growth was monitored for one week. The total accumulated UV doses for 24-hour periods were 20, 15, 10, and 0 (control) mJ/cm$^2$, respectively. Under non-irradiated control conditions, the spots increased their diameter by 3.28 ± 0.96 mm after seven days (Fig 2B). In contrast, the lowest UV dose duty cycles (10 and 15 mJ/cm$^2$) showed only 0.53 ± 0.42 mm and 0.25 ± 0.30 mm increases in diameter after seven days, respectively. For 20 mJ/cm$^2$, no growth was observed.

The most rapid growth increase occurred during the first five days, and representative pictures of the growth development are presented in Fig 2C. In the non-irradiated control condition, hyphae stretched out around the seeded spot, along with vertical growth and a denser mycelium after three to five days. Although there was a slight increase in the measured diameter during the 10 and 15 mJ/cm$^2$ conditions (Fig 2B), visual assessment (Fig 2C) showed that the growth appeared markedly different, lacking both a hyphae circle around the spot and visible vertical growth. No change was observed in the 20 mJ/cm$^2$ condition. This result suggests that the growth of *P. candidum* is inhibited by far-UVC light at doses below the TLV of 23 mJ/cm$^2$.

## Discussion

This study aimed to assess the inhibiting effect of far-UVC light applied in low-dose duty cycles on growing mold, and we found that the light significantly inhibited the growth of *P. candidum*.

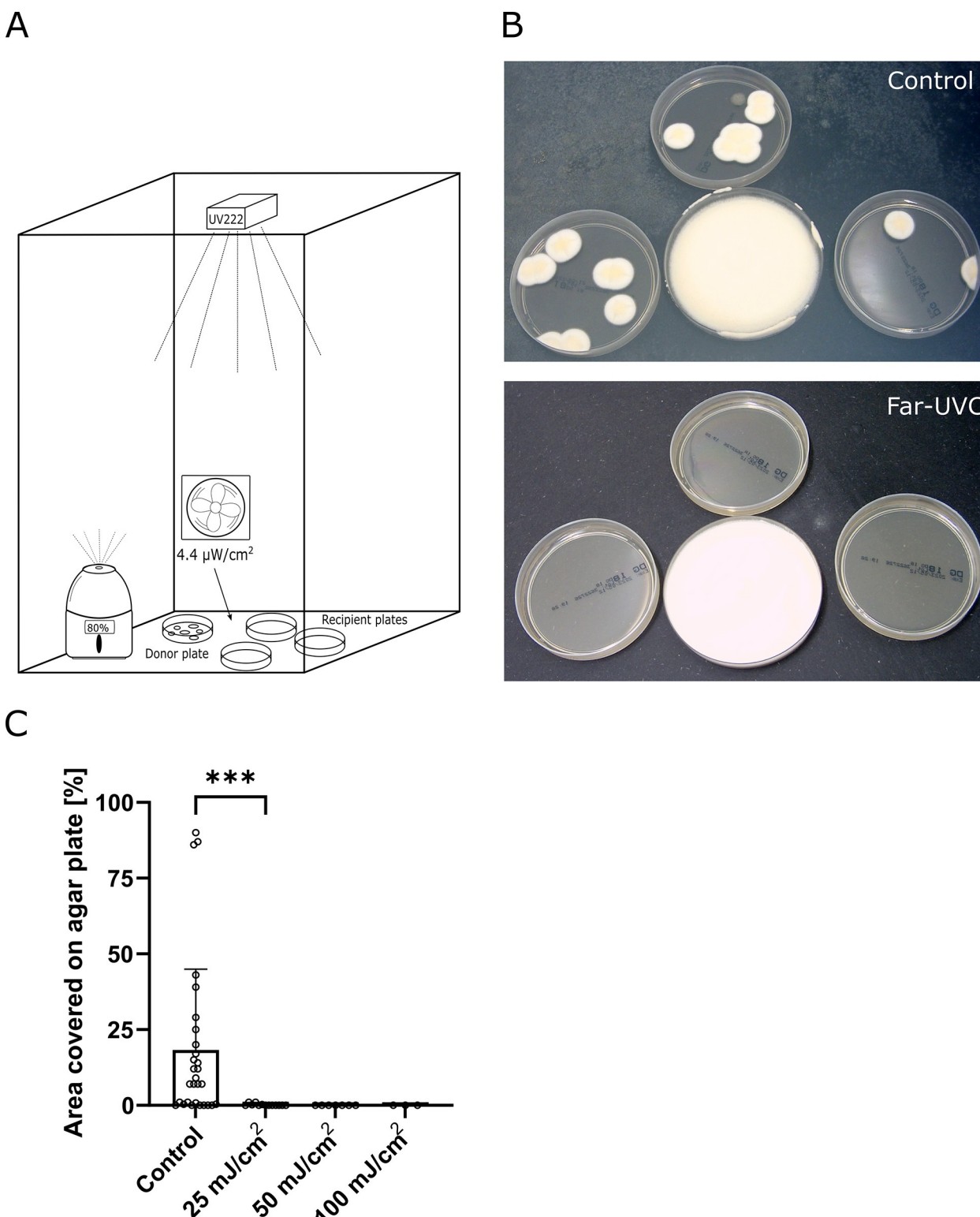

**Fig 1. Far-UVC inhibits *P. candidum* growth induced by airborne spores.** A donor plate covered with *P. candidum* was placed in a box with 80% humidity, air movement, and a far-UVC lamp, and recipient plates were positioned adjacently to the donor plate (A). Accumulated doses of cycling far-UVC light were 25, 50, and 100 mJ/cm² per 24 hours. Representative pictures show fungal growth on the recipient agar plates with and without far-UVC light (B). The growth was monitored by measuring the percentage of the agar surface covered by mycelium (C). The mean percentage values for surface coverage were 21.7% (control), <1% (25 mJ/cm²), 0% (50 mJ/cm²), and 0% (100 mJ/cm²). Statistical analysis was performed using

unpaired Welch's t-test, with the asterisks indicating a p-value of 0.0005. The p-values for 50, and 100 mJ/cm$^2$ conditions versus control were both 0.0005 (not shown in the figure). Columns indicate mean values with standard deviation, and each dot represents a recipient agar plate. The results indicate that cycling far-UVC light can inhibit mold growth, even when the dose is only 25 mJ/cm$^2$ per 24 hours.

UVC light emitted from conventional mercury lamps is well-known for its antimicrobial properties, but its use is limited to areas without people present due to health hazards, including the risk of causing skin and eye damage [32]. Consequently, the conventional UVC light is typically administered in one large dose while no people are present or used for upper-room decontamination. Most knowledge of UVC light applied for decontamination is based on a century of use of this method. The development of excimer lamps emitting far-UVC light at 222 nm, which is safe for human exposure as described in the introduction, has opened doors to a range of new applications. Currently, we are learning how to use far-UVC light for decontamination while people are present, ensuring that the TLV is never exceeded. This involves delivering the light in intermittent low doses throughout the day. However, most studies are based on applying single large doses [12], as was customary when applying conventional UVC light, typically at 254 nm.

In this study, we found that intermittent cycling doses can effectively inhibit growth, even at low doses below the TLV of 23 mJ/cm$^2$. These results suggest that far-UVC light can be a valuable tool in environments prone to mold attacks and could be used as a chemical-free supplement to manual cleaning.

A study of 16,190 people in Denmark, Estonia, Iceland, Norway, and Sweden revealed a prevalence of damp indoor environments of 18%, with a higher occurrence of respiratory symptoms and asthma among those living in damp housing [33]. Another study estimated that approximately 50% of residents of homes in the USA are at higher risk of experiencing respiratory symptoms due to exposure to dampness and/or mold in their homes [34]. The risk was not limited to homes but extended to schools, offices, and other institutional buildings. In hospitals, mold outbreaks can cause fungal infections in vulnerable and immunocompromised patients without legitimate clinical explanations [10, 35, 36]. Fungi-related health hazards primarily stem from airborne spores polluting the air. This study does not address the effect of far-UVC light on airborne spores directly, and there is a probability that the doses used here are too low to have a significant effect on the spores. Previously, the log 1 reduction dose for *Penicillium expansum* spores has been reported to be 11 mJ/cm$^2$ [29], applied as a single large dose. However, as indoor mold growth increases the risk of high concentrations of airborne spores, preventing growth on indoor surfaces is crucial to securing indoor air quality [11]. When the TLV is increased from 23 mJ/cm$^2$ to 161 and 479 mJ/cm$^2$ for eyes and skin, respectively, an enhanced inactivating effect on fungi is most likely achievable. Further studies on far-UVC light in real-world settings should be pursued to investigate if the light can enhance indoor air quality and prevent some of the symptoms associated with poor air quality.

## Conclusion

Indoor damp environments are particularly susceptible to mold infestations, which can substantially decrease indoor air quality and trigger adverse health effects such as asthma and allergies, or cause life threatening diseases in vulnerable hospital patients. To protect public health, indoor mold outbreaks should be prevented. In this study, we investigated the inhibitory impact of far-UVC light on the growth of *P. candidum* mold. Our findings suggest that cycling far-UVC doses below the TLV of 23 mJ/cm$^2$ can effectively inhibit mold growth. This result emphasizes the potential of far-UVC light at 222 nm in protecting indoor environments and public health by inhibiting mold growth on indoor surfaces.

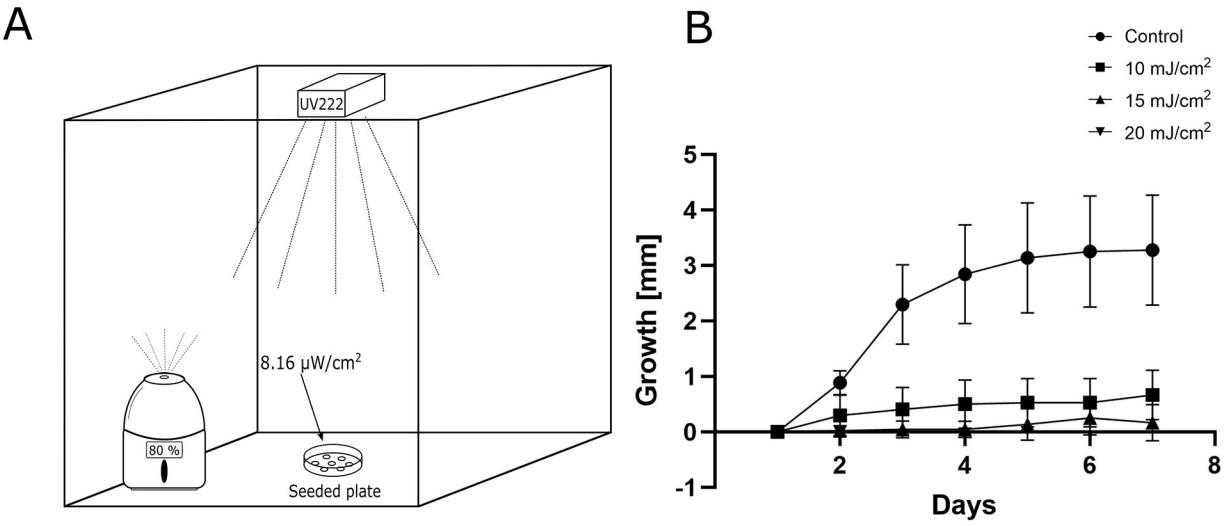

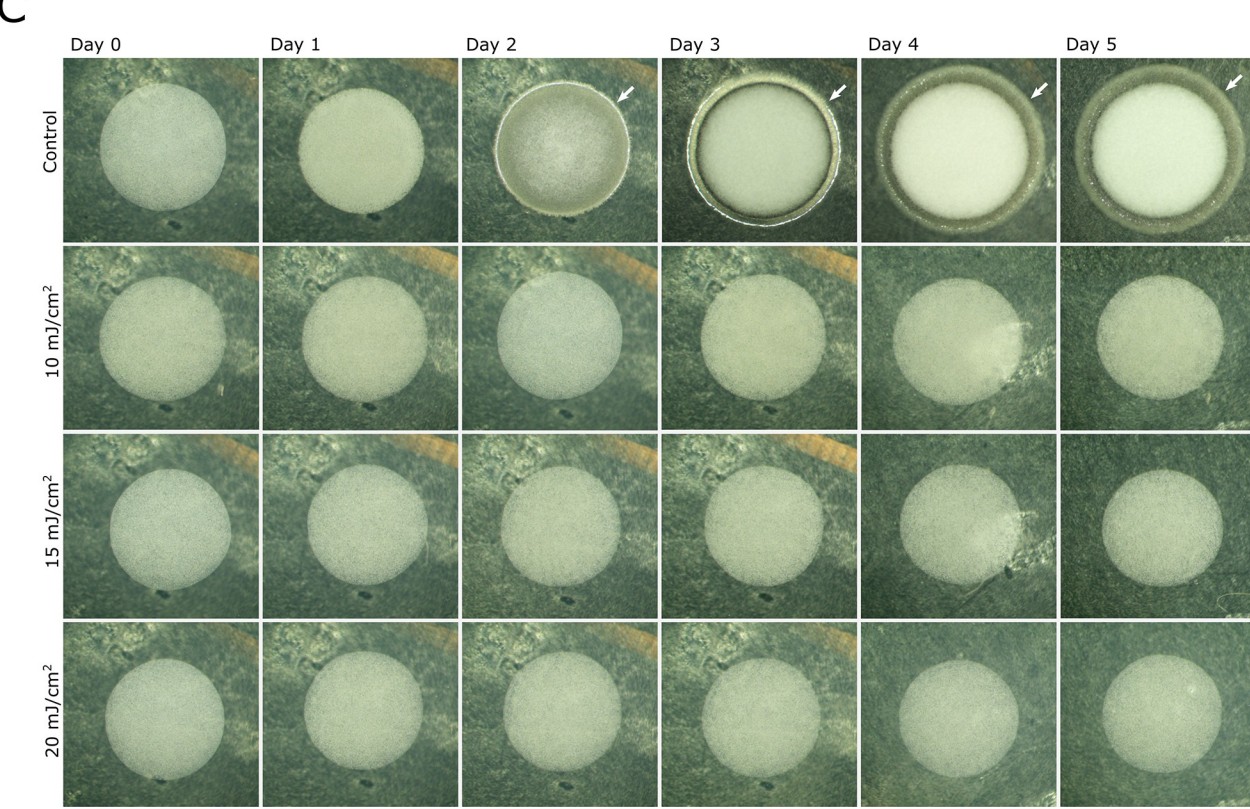

**Fig 2. Far-UVC inhibits both vertical and horizontal growth of *P. candidum*.** *P. candidum* spores were seeded onto agar plates and left to grow in 80–90% humidity for seven days while being exposed to far-UVC light (A). Growth was monitored daily by measuring diameter increases (B, C) and through visual assessment (C). After seven days, the mean diameter increases were $3.28 \pm 0.96$, $0.53 \pm 0.42$, $0.25 \pm 0.30$ and $0.0$ mm for control, 10, 15, and 20 mJ/cm$^2$, respectively. Growth in the non-irradiated control condition was significantly higher (Welch's unpaired t-test p-value $<0.001$) compared to any of the irradiated samples. Representative images (C) show notable hyphal growth (indicated by white arrows) from day two under non-irradiated control conditions, which is absent in all the of the far-UVC irradiated samples. This indicates that cycling doses of far-UVC below the TLV inhibit the growth of *P. candidum*.

## Supporting information

**S1 Fig. *Penicillium candidum* growth on pre-irradiated agar plates.** DG18 agar plates were irradiated with either 0, 100 or 400 mJ/cm$^2$ far-UVC light. Immediately after irradiation, P. candidum spores ($3.4 \times 10^6$ spores/mL) was seeded on the plates in 8 μL droplets with five droplets on each plate. The inoculated plates were incubated at 24°C for three days, and the diameter of the seeded droplets was measured before and after incubation. The procedure was repeated twice. The growth on the plates was found to be not significantly different (one-way ANOVA, p > 0.05), indicating that the DG18 agar plates were not affected by far-UVC irradiation.
(DOCX)

**S2 Fig. Measurements of ozone generated from the far-UVC lamp.** Ozone measurements were conducted in the boxes using the same setup described in the methods section "Inhibition of *P. candidum* growth from airborne spores by far-UVC light", with the highest UV dose of 100 mJ/cm$^2$ per 24 hours. Ventilation was on, and the humidity was set to 80%. A calibrated EZ-1Z (Scanion, Denmark), which measures ozone between 20–140 ppb, was placed at the bottom of the box in the same location as the agar plates. After reaching equilibrium, the ozone level was below the detection limit. As a positive control, the lamp was configured to be constantly on, resulting in the minimum detectable level of 20 ppb. The blue graph represents the ozone level when the lamp is always on, while the red graph represents the theoretical oscillating ozone levels during the duty cycle configuration.
(DOCX)

## Acknowledgments

The authors would like to thank Peter Tønning for his assistance with ozone measurements.

## Author Contributions

**Conceptualization:** Emilie Hage Mogensen, Christian Kanstrup Holm.

**Data curation:** Emilie Hage Mogensen.

**Formal analysis:** Emilie Hage Mogensen.

**Investigation:** Emilie Hage Mogensen.

**Methodology:** Emilie Hage Mogensen.

**Supervision:** Christian Kanstrup Holm.

**Writing – original draft:** Emilie Hage Mogensen.

**Writing – review & editing:** Emilie Hage Mogensen, Christian Kanstrup Holm.

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
