## [Decision Letter · Decision Letter 0]

17 Apr 2024

PONE-D-24-05537Intermittent low-intensity far-UVC irradiation inhibits growth of common mold below threshold limit valuePLOS ONE

Dear Dr. Mogensen,

Thank you for submitting your manuscript to PLOS ONE. After careful consideration, we feel that it has merit but does not fully meet PLOS ONE’s publication criteria as it currently stands. Therefore, we invite you to submit a revised version of the manuscript that addresses the points raised during the review process.

We look forward to receiving your revised manuscript.

Kind regards,

Rajeev Singh

Academic Editor

PLOS ONE

Journal Requirements:

EHM and CKH received salary from UV Medico A/S.

I have read the journal's policy and the authors of this manuscript have the following competing interests: EHM is a full-time employee and CKH is a co-founder and an employee at UV Medico A/S.

We note that one or more of the authors are employed by a commercial company: UV Medico A/S. 

“The funder provided support in the form of salaries for authors, but did not have any additional role in the study design, data collection and analysis, decision to publish, or preparation of the manuscript. The specific roles of these authors are articulated in the ‘author contributions’ section.”

4. We note that your Data Availability Statement is currently as follows: All relevant data are within the manuscript

If there are ethical or legal restrictions on sharing a de-identified data set, please explain them in detail (e.g., data contain potentially sensitive information, data are owned by a third-party organization, etc.) and who has imposed them (e.g., an ethics committee). Please also provide contact information for a data access committee, ethics committee, or other institutional body to which data requests may be sent. If data are owned by a third party, please indicate how others may request data access.. 

Reviewers' comments:

Reviewer's Responses to Questions

**Comments to the Author**

1. Is the manuscript technically sound, and do the data support the conclusions?

Reviewer #1: Yes

Reviewer #2: Yes

2. Has the statistical analysis been performed appropriately and rigorously? 

Reviewer #1: Yes

Reviewer #2: Yes

3. Have the authors made all data underlying the findings in their manuscript fully available?

Reviewer #1: Yes

Reviewer #2: Yes

4. Is the manuscript presented in an intelligible fashion and written in standard English?

Reviewer #1: Yes

Reviewer #2: Yes

5. Review Comments to the Author

**Reviewer #1:** The authors, who work for a manufacturer of Far-UVC lamps, investigated the antimicrobial effect of Far-UVC radiation on the fungus Penicillium candidum. In principle, this is a meaningful study, as fungi cause millions of infections every year. 222 nm far-UVC has the advantage over conventional 254 nm UVC radiation from mercury vapor lamps that this radiation is presumably relatively harmless to humans and therefore far-UVC radiation sources could be operated in the presence of humans.

Some important details are missing, but otherwise the article is easy to understand and provides a statement with novelty value if the points of criticism raised can be clarified.

Important questions:

• Why was this fungus (species) selected, which presumably has no medical significance?

• Why was the irradiation carried out on agar plates? How can it be ensured that the antimicrobial effect was not influenced by ROS formation in the agar? (See e.g., https://doi.org/10.1016/s0891-5849(01)00545-7 )

• Was the ozone concentration in the boxes measured? Ozone might also have influenced your results.

• The experimental data on airborne spores show the success of irradiation with a dose of 25 mJ/cm2. However, only 23 mJ/cm2 per day is permitted in Europe. Why were no experiments carried out with 23 mJ/cm2? The statement in the title of paper is only verified to a limited extent - even if the difference in dose is less than 10%.

Minor comments:

• The reference for the 23 mJ/cm2 threshold seems to be missing? (The commission is mentioned in the text but seems to be missing in the References?)

• It should be pointed out that the limit values mentioned apply to healthy adults. There have been no studies on children or (skin) patients to date.

• How was the irradiance measured?

• More detailed information on the fungus? (Which strain?)

• How often were the experiments repeated for the air borne spores?

• How many spores are 2 mg/ml?

• Why is the background in Figure 2C so green?

• The illustration in Figure 2C is not very clear. Visually, for example, it looks as if the fungal lawn in the non-irradiated control was less on day 5 than on day 4? For all other irradiation intensities, I cannot see any difference between the agar plates.

• The mentioned dose from [27] is wrong. (254 nm value instead of 222 nm dose).

**Reviewer #2:** This is well written paper. It would be good to see it published. The experiments were carried out in a scientific method and the paper can be submitted for more technical reviews in scientific journals.

6. PLOS authors have the option to publish the peer review history of their article (what does this mean?). If published, this will include your full peer review and any attached files.

Reviewer #1: **Yes: **Martin Hessling

Reviewer #2: **Yes: **Rajul Randive

---

## [Author Response · Author response to Decision Letter 0]

3 Jun 2024

Dear editor and reviewers

Thank you for giving your time and expertise to review our manuscript now entitled “Intermittent low-dose far-UVC irradiation inhibits growth of common mold below threshold limit value”. 

In the revised manuscript, we have carefully considered reviewers’ comments and suggestions, and we have replied to each comment in a point-by-point fashion. We appreciate all your input and valuable comments, which definitely have improved the quality of our manuscript. 

Please find below all our responses. 

Sincerely

Emilie Hage Mogensen

 

Reviewer #1: The authors, who work for a manufacturer of Far-UVC lamps, investigated the antimicrobial effect of Far-UVC radiation on the fungus Penicillium candidum. In principle, this is a meaningful study, as fungi cause millions of infections every year. 222 nm far-UVC has the advantage over conventional 254 nm UVC radiation from mercury vapor lamps that this radiation is presumably relatively harmless to humans and therefore far-UVC radiation sources could be operated in the presence of humans.

Some important details are missing, but otherwise the article is easy to understand and provides a statement with novelty value if the points of criticism raised can be clarified.

Thank you very much. 

Important questions:

• Why was this fungus (species) selected, which presumably has no medical significance?

Thank you for your question. We chose Penicillium because it is one of the most common species found in water-damages buildings. We agree that it would be interesting to conduct studies with species of greater medical importance. In fact, we are currently working on projects with both Aspergillus spp. and Candida auris, but the data is not ready for publication. 

• Why was the irradiation carried out on agar plates? How can it be ensured that the antimicrobial effect was not influenced by ROS formation in the agar? (See e.g., https://doi.org/10.1016/s0891-5849(01)00545-7 )

Thank you for raising this question. The experiment was also carried out on other surfaces including wood and bread. We observed the same result every time with little to no growth when far-UVC was applied. However, as the colonies grew very unevenly on these varying surfaces, it was difficult to measure the growth. The smooth surfaces of the agar plates were more suitable. 

To further address this question, we have performed an additional experiment. Briefly, we exposed DG18 agar plates to 100 and 400 mJ/cm2, which corresponds to the highest dose accumulated over 24 hours and over 4 days, respectively. The 400 mJ/cm2 dose was chosen because this is the approximate dose that has accumulated when the first colonies, stemming from airborne spores, started to grow. 

After irradiation, P. candidum spore-solution was seeded onto the plates in 8 µL droplets and incubated at 24 degrees for 72 hours. As control, the spore solution was seeded onto non-irradiated DG18 agar plates. After 72 hours, the growth was measured as diameter increase. 

We found no significant growth difference in the 100, 400 mJ/cm2 or non-irradiated control condition. 

The data has been added to Supplementary Information (S1) and mentioned in the result section.

• Was the ozone concentration in the boxes measured? Ozone might also have influenced your results.

We appreciate your insightful suggestion. As requested, ozone concentrations have now been measured in the boxes for the highest accumulated dose of 100 mJ/cm2 per 24 hours. We repeated the setup with both ventilation and 80% humidity as described in the methods section. An apparatus for measuring ozone was placed in the same location as the agar plates. The apparatus was a calibrated EZ-1X (Scanion, Denmark), which measures ozone between 0.02-0.14 ppm. We recorded the ozone levels over three days with the lamp on and three days with the lamp off. All measurements were below the detection limit of 0.02, demonstrating that it is unlikely that ozone affected the results on the P. candidum growth. The data has been added to Supplementary Information (S2) and mentioned in the result section. 

• The experimental data on airborne spores show the success of irradiation with a dose of 25 mJ/cm2. However, only 23 mJ/cm2 per day is permitted in Europe. Why were no experiments carried out with 23 mJ/cm2? The statement in the title of paper is only verified to a limited extent - even if the difference in dose is less than 10%.

Thank you for your question. In our study, the first experiment applied a minimum dose of 25 mJ/cm2, slightly above the 23 mJ/cm2 threshold limit value. However, lower doses are examined in the second part of the paper, where we conducted experiments with seeded spores on agar plates using doses of 0, 10, 15, and 20 mJ/cm2. We observed that a dose as low as 10 mJ/cm2 inhibited growth. Therefore, the title of the paper reflects the combined results from both parts of the study, not just the experiment with airborne spores. We hope this clarification addresses your concerns about the title.

Minor comments:

• The reference for the 23 mJ/cm2 threshold seems to be missing? (The commission is mentioned in the text but seems to be missing in the References?)

Thank you for making us aware of this mistake. The references have now been added in the Introduction. 

• It should be pointed out that the limit values mentioned apply to healthy adults. There have been no studies on children or (skin) patients to date.

As requested, this has now been pointed out in the Introduction. We appreciate your insightful suggestion. 

• How was the irradiance measured?

As requested, this has been added in the methods section. We used a handheld UIT 2400 light meter from Ushio Inc., Japan. 

• More detailed information on the fungus? (Which strain?)

As requested, we have added this information in the methods section. We do not know the exact strain, but we have included product details and manufacturer (Chr. Hansen, Denmark). 

• How often were the experiments repeated for the air borne spores?

As requested, this is now clarified in the methods section. We already stated that the lowest dose of 25 mJ/cm^2, which is the most interesting because it’s close to the TLV of 23 mJ/cm^2, was repeated with four technical replicates, but we have now mentioned that the doses above 25 mJ/cm^2 were screened once. 

• How many spores are 2 mg/ml?

As requested, this has now been counted and added to the Methods section. The spore concentration was 3.4 × 106 spores/mL. 

• Why is the background in Figure 2C so green?

Thank you for your question. The background surface is coincidentally green and can be seen through the transparent agar. 

• The illustration in Figure 2C is not very clear. Visually, for example, it looks as if the fungal lawn in the non-irradiated control was less on day 5 than on day 4? For all other irradiation intensities, I cannot see any difference between the agar plates.

Thank you making us aware of this. As requested, we have updated the figure for clarity. The apparent smaller size of the non-irradiated control on day 5 compared to day 4 was due to a slight scaling error in the image; however, this does not affect the data. Measurements were taken manually with a ruler directly on the agar plates, not from the pictures. The images serve to visually document growth, particularly the vertical hyphae, which are visible to the naked eye but challenging to measure directly. To avoid confusion, we have removed the diameter labels from the images and instead added arrows to highlight visible growth that occurs only in the non-irradiated control. This visually supports our findings that very low doses of far-UVC effectively inhibit the growth of P. candidum.

• The mentioned dose from [27] is wrong. (254 nm value instead of 222 nm dose).

Thank you for this comment. Maybe we have misunderstood this comment, but to us it is unclear what the comment is referring to. Reference [27] is a paper by Clauss et al., 2006 with the title “Higher effectiveness of photoinactivation of bacterial spores, UV resistant vegetative bacteria and mold spores with 222 nm compared to 254 nm wavelength”. This paper assessed both 254 and 222 nm inactivating doses on microorganisms including fungi. 

We would appreciate more details to be able to properly reply to this comment. 

Reviewer #2: This is well written paper. It would be good to see it published. The experiments were carried out in a scientific method and the paper can be submitted for more technical reviews in scientific journals

Thank you very much. 

Thanks to editor and reviewers for constructive feedback and comments.

---

## [Editor Report · Decision Letter 1]

10 Jun 2024

Intermittent low-dose far-UVC irradiation inhibits growth of common mold below threshold limit value

PONE-D-24-05537R1

Dear Dr. Mogensen,

We’re pleased to inform you that your manuscript has been judged scientifically suitable for publication and will be formally accepted for publication once it meets all outstanding technical requirements.

Kind regards,

Rajeev Singh

Academic Editor

PLOS ONE
---

## [Editor Report · Acceptance letter]

24 Jun 2024

PONE-D-24-05537R1 

PLOS ONE

Dear Dr. Mogensen, 

I'm pleased to inform you that your manuscript has been deemed suitable for publication in PLOS ONE. Congratulations! Your manuscript is now being handed over to our production team.

Kind regards, 

on behalf of

Dr. Rajeev Singh 

Academic Editor

PLOS ONE